# Clinical Presentation and Novel Pathogenic Variants among 68 Chinese Neurofibromatosis 1 Children

**DOI:** 10.3390/genes10110847

**Published:** 2019-10-26

**Authors:** Ruen Yao, Tingting Yu, Yufei Xu, Li Yu, Jiwen Wang, Xiumin Wang, Jian Wang, Yiping Shen

**Affiliations:** 1Department of Medical Genetics and Molecular Diagnostic Laboratory, Shanghai Children’s Medical Center—Shanghai Jiao Tong University School of Medicine, Shanghai 200127, China; yaoruen@126.com (R.Y.); yutingting@scmc.com.cn (T.Y.); xuyufei82@sjtu.edu.cn (Y.X.); veronica_yuli@163.com (L.Y.); wangjian@scmc.com.cn (J.W.); 2Department of Pediatrics, Shanghai Children’s Medical Center—Shanghai Jiao Tong University School of Medicine, Shanghai 200127, China; wangjiwen@scmc.com.cn (J.W.); wangxiumin@scmc.com.cn (X.W.); 3Division of Genetics and Genomics, Boston Children’s Hospital, Boston, MA 02115, USA; 4Department of Neurology, Harvard Medical School, Boston, MA 02115, USA

**Keywords:** neurofibromatosis 1, café-au-lait macules, novel variants, exome, short stature

## Abstract

Background: Neurofibromatosis 1 (NF1) is one of the most common dominantly inherited genetic disorders worldwide, with an age-dependent phenotypic expression. Exploring the mutational spectrum and clinical presentation of NF1 patients at different ages from a diverse population will aid the understanding of genotype–phenotype correlations. Methods: In this study, 95 Chinese children with clinical suspicion of NF1 mainly due to the presence of multiple café-au-lait macules (CALMs) were subjected to medical exome-sequencing analysis and Sanger confirmation of pathogenic variants. Clinical presentations were evaluated regarding dermatological, ocular, neurological, and behavioral features. Results: Pathogenic or likely pathogenic *NF1* variants were detected in 71.6% (68/95) of patients; 20 pathogenic variants were not previously reported, indicating that Chinese NF1 patients are still understudied. Parental Sanger sequencing confirmation revealed 77.9% of de novo variants, a percentage that was much higher than expected. The presence of a higher number of NF1-related features at young ages was correlated with positive diagnostic findings. In addition to CALMs, neurological and behavioral features had a high expression among Chinese NF1 children. We attempted to correlate short stature with the locations of the pathogenic variants across the *NF1* gene. It is interesting to notice that variants detected in the C-terminal region of the *NF1* gene were less likely to be associated with short stature among the NF1 patients, whereas variants at the N-terminal were highly penetrant for the short stature phenotype. Conclusion: Novel *NF1* pathogenic variants are yet to be uncovered in under-studied NF1 patient populations; their identification will help to reveal novel genotype–phenotype correlations.

## 1. Introduction

Neurofibromatosis 1 (NF1) is one of the most common dominantly inherited genetic disorders with a worldwide incidence of at least one in 3000 individuals [1,2,3]. The protein product of the *NF1* gene is neurofibromin, which acts as a tumor suppressor protein regulating the Ras signaling pathway, thus NF1 has been included in the group of rare genetic conditions called Rasopathies [4]. Superficial features such as multiple café-au-lait macules (CALMs), axillary and inguinal freckling, multiple cutaneous neurofibromas, and iris Lisch nodules are often the reasons to bring individuals for medical attention. In particular, CALMs often appear early in life and become the initial indication of NF1. Yet, most NF1-related features exhibit an age-dependent expression. For children with a suspicion of NF1 who do not yet meet the clinical diagnostic criteria at an early age, molecular testing of NF1 is not only helpful to confirm the diagnosis but also beneficial for differential diagnosis between Rasopathies and other diseases sharing overlapping features. The diagnostic yield of NF1 is very high for adults and for those who meet the clinical diagnostic criteria, whereas the yield is somewhat lower for children who do not present with a typical pattern of NF1 but are suspected of NF1 by presenting CALMs. The diagnostic yield for children would vary from one center to another depending on the clinical experience and the judgments of the clinicians. We had previous reported that 13 out of 19 Chinese children with CALMs were molecularly confirmed to have NF1 at the Shanghai Children’s medical center (SCMC)—Shanghai Jiaotong University School of Medicine [5]. We have since expanded the number of patients examined.

Over 3000 NF1 pathogenic variants have been reported, but the pathogenic variants from Chinese NF1 patients are under-represented [6]. Exploring novel pathogenic variants and their distribution in a diverse population may help to better understand genotype–phenotype correlations which are still quite limited [7]. Previous works have focused on dissecting the mutation spectrum and clinical features of the NF1 Chinese population [8,9,10]. Here, we planned to perform a large-scale genotype–phenotype study of Chinese NF1 children and we report the first set of findings for 95 Chinese children from a single center.

## 2. Material and Methods

### 2.1. Patients

Individuals with clinical suspicion of NF1, mostly due to the presence of multiple CALMs, and patients clinically diagnosed with NF1 but not molecularly confirmed were referred to the Department of Medical Genetics at SCMC—Shanghai Jiaotong University School of Medicine (Shanghai, China), between January 2017 and May 2019. A total of 95 unrelated individuals (54 males, 41 females) were included in this study. The age of the patients ranged from 6 days to 17 years, with an average age of 6 years and 2 months. Ethical approval for this study was obtained from the Ethics committee of SCMC—Shanghai Jiaotong University School of Medicine (Project Identification Code: SCMCIRB-Y2019021; Date of Approval 25 February 2019). Written informed consent was obtained from the proband’s parents. Patients’ medical chart review was performed for information related to dermatological, ocular, neurologic, musculoskeletal, cardiac, and metabolic features.

### 2.2. Molecular Genetic Analysis

Medical exome sequencing was completed at the molecular diagnostic laboratory of SCMC. Peripheral blood samples were collected from the patients and their parents. Targeted capture of medical exome was performed using Agilent SureSelect capture kits (Agilent, Santa Clara, CA, USA). The captured libraries were sequenced using the Illumina HiSeq 2500 system, and base-calling and sequence-read quality assessment were performed using Illumina HCS 2.2.58 software (Illumina, San Diego, CA, USA). Sequencing data with sufficient on-target coverage were aligned to hg19 using BWA-MEM, and variant calling was performed with Genome Analysis Toolkit. All single-nucleotide variants and indels were saved as VCF format files and annotated with both Ingenuity^®^ Variant Analysis™ (Ingenuity Systems, Redwood City, CA, USA) and TGex (Translational Genomic Expert) for variation filtering and interpretation. The transcript NM_000267.3 was used for *NF1* gene variant annotation. All the exonic and splicing variants including possible mosaicism variants in the medical exome we captured were classified according to the recommended method of the American College of Medical Genetics and Genomics [11]. Pathogenic and potentially pathogenic variants were confirmed by Sanger sequencing and validated by parental testing.

## 3. Results

### 3.1. Genetic Findings

Pathogenic or likely pathogenic variants were detected in 71.6% of patients (68/95), including 35 nonsense variants, 19 missense variants, 12 slicing variants, and 2 in-frame deletions.

All *NF1* variants were confirmed by Sanger sequencing in the probands and their parents. Fifty-four variants were confirmed as de novo (77.9%); among those, 70.3% (38/54) were null variants, and the rest 29.7% (16/54) were missense variants or in-frame deletions. All individuals with a clear family history and CALMs had positive NF1 testing results. In addition, nine variants were maternally inherited (13.2%), and five variants were paternally inherited (35.7%); among those inherited, 64.3% (9/14) were null variants, and the rest 35.7% (5/14) were missense variants. We detected 20 novel variants among this cohort of Chinese NF1 patients (Table 1).

The diagnostic yield gradually increased in different age groups (Figure 1), indicating that clinical diagnosis improved for older children. The lowest yield was 61.54% (16/26) for the infant and toddler group. It improved to 67.86% (19/28) and 77.14% (27/35) for the preschooler and school-age groups, respectively. Six adolescent patients were confirmed by molecular testing.

### 3.2. Clinical Characteristics

The clinical characteristics of 68 individuals with pathogenic *NF1* variants and 27 individuals without NF1 findings were analyzed (Table 2 and Appendix A). Multiple CALMs were the initial indication for the suspicion of NF1 and were observed in all 95 individuals. We found that 86.8% (59/68) of molecularly confirmed NF1 patients showed CALMs at birth. By the age of 18 months, all NF1 patients exhibited CALMs. Three patients with NF1 pathogenic variants presented with less than six CALMs and, thus, did not meet the clinical diagnostic criteria [12], whereas for those without molecular confirmation, two-thirds of them exhibited CALMs after birth, and the CALMs were mostly atypical, smaller in size, greater in number, and scattered all over the body. Cutaneous neurofibromas were observed in 13/68 (19.11%) NF1 patients, whereas plexiform neurofibromas were rarely observed in this cohort of Chinese NF1 patients. MRI was not routinely performed in most patients, unless plexiform neurofibromas or optic gliomas were observed or suspected.

Optic atrophy was noticed in a five-year-old boy with a NF1 pathogenic variant. Most patients did not receive formal ophthalmological examination, so the prevalence of Lisch nodules was unknown. Some obvious eye phenotypes such as myopia, astigmia, and strabismus were noticed.

Among patients with *NF1* pathogenic variants, 31/68 (45.59%) had varying degrees of neurological manifestations, which included attention deficit and hyperactivity (ADHD), developmental delay, language deficit, sleep disorders, learning problems, intellectual disability, and headache of unknown cause. Male patients seemed more likely to have neurological problems compared with female patients (50% (19/38) males and 40% (12/30) females). A third (23/68) of NF1 children (13 males and 10 females) exhibited short stature. Six female patients exhibited sexual precocity during adolescence. Musculoskeletal features and cardiac issues were rarely identified in this cohort. Only one patient had scoliosis. One female patient had an atrial septal defect.

The prevalence of the NF1-related clinical features was plotted for each age groups (Figure 1). Typical CALMs were highly present in all four age groups. Over 60% of school-age children had neurological problems. Cutaneous neurofibromas were not detected in any patients under two years of age but were much more prevalent in preschool children (3–6 years old) and adolescents (14–18 years old). Age-dependent increased prevalence of features was observed in general, which correlated with the molecular diagnostic yield.

### 3.3. Variant Distribution and Short Stature

To explore a possible correlation between NF1 pathogenic variants and short stature, we plotted the variants’ distribution along the *NF1* gene (Figure 2). We found that 56.5% (13/23) of the individuals exhibited short stature when their pathogenic variants were located in the 5′ tertile part of the *NF1* gene (residues 1–909 including cysteine/serine-rich domain (CSRD)), whereas the percentage of short stature decreased to 39.1% and 4.3% for variants’ locations in the middle tertile part (residues 910–1818, including the GTPase-activating protein-related domain (GRD) and Sec14-like domain (Sec14)) and in the 3′ tertile part (residues 1819–2818, including the focal adhesion kinase domain (FAK)). Since the majority of detected variants are loss-of-function variants which include nonsense, frameshift, and splicing variants, we evaluated the percentage of such variants in the short-stature cohort. While 42.1% (8/19) of NF1 patients with loss-of-function variants in the 5′ tertile part had short stature, two patients (13.3%, 2/15) with loss-of-function variants in the middle tertile part and only one patient (16.7%) with a loss-of-function variant in the 3′ tertile part had short stature.

## 4. Discussion

Since NF1 prevalence is consistent across different ethnic groups, there should be over 400,000 NF1 patients in China. However, among the over 3000 published *NF1* pathogenic variants, few were initially reported in Chinese NF1 patients. As a consequence, we still do not have a good understanding of the genetic profile of the Chinese NF1 population. Our study represents one such effort towards a better understanding of the genotype and phenotype of Chinese NF1 patients. The current study involved data only from a single center but it is so far the largest one. The finding of a significant number of novel pathogenic variants in this cohort indicates that the Chinese NF1 patient population is still under-studied. The findings of unique genomic profile and a potential novel genotype–phenotype correlation between short stature and variant location suggest that such effort is necessary and meaningful towards a more complete understanding of this condition worldwide. We will continue to conduct large-scale NF1 genotype and phenotype studies among the Chinese NF1 patient population.

The *NF1* gene is one of the highly mutable genes in human genome involved in dominant disorders. It is well documented that about half of the affected individuals have de novo *NF1* mutations and the other half have familial variants. It is surprising to note that the percentage of de novo variants (77.9%) detected in this cohort was much higher than expected. We offer the following explanations for this finding. We think that a combination of increasingly available genetic diagnostic services and prenatal diagnostic testing in China could have played a role in reducing the birth rate of affected fetuses or preventing the passing of the familial pathogenic variants in the first place via pre-implantation genetic testing. While this is possibly a significant factor, we are not sure if this could explain all the cases. In addition, we think it is also possible that individuals with de novo pathogenic variants are more likely to seek medical attention because of the presence of features not seen in family members. Whether the higher-than-expected de novo rate in this cohort is due to patients’ Chinese ethnicity will need to be studied with a large dataset.

Although the medical exome approach we used can help the differential diagnosis of NF1-like syndromes such as Legius syndrome and the detection of mosaic variants at some levels, the missed detections of small deletions, low-level mosaic mutations, and deep-intronic variants contributed to our overall (71.6%) lower-than-typical level of diagnostic yield. However, this yield also reflect the uncertainty of NF1 clinical diagnosis among children. CALMs are usually the first clinical indication for suspected NF1. Typically, the characteristic CALMs in individuals with NF1 are ovoid in shape, with well-defined borders, uniform in color, and about 1–3 cm in size [13]. Genetically confirmed NF1 children presented with typical CALMs in this study and in our previous study [5]. Cutaneous neurofibromas developed in our cohort after two years of age, so the molecular diagnostic testing is more important for early confirmation of those who only present CALMs. Yet, a comprehensive genetic testing is required, especially in patients with typical CALMs, to improve the diagnostic efficiency [14,15]. We observed an increasing molecular diagnostic yield in older age groups of patients, which was consistent with the increasing prevalence of NF1-related features and, thus, with a better accuracy of clinical diagnosis of NF1 in old-age children.

More than half of the NF1 children exhibited neurological and behavioral issues including intellectual/learning disability and ADHD in our cohort, which is consistent with findings in NF1 patients from other countries [16,17]. Intellectual disability is often seen in 6–7% of NF1 patients, a frequency about twice that of the general population [18]. A slightly higher percentage of 8.8% (6/58) of individuals in our cohort exhibited intellectual disability and developmental delay. Neurological and behavioral problems were mostly present among school-aged children. In addition, three patients had headache, and four patients had sleep disorders, which are also known to be common in NF1 patients.

Individuals with NF1 tend to be below-average in height [19]. In a study, 15% of NF1 patients were reported to have a height below the 3rd percentile [20]. In our cohort, 23 of 68 (33.8%) molecularly confirm NF1 children had a height below the 3rd percentile. This is the first such data based on Chinese NF1 patients from a single center. We will continue to examine the growth characteristics of Chinese NF1 patients. Interestingly, NF1 variants in patients with short stature were more likely to be distributed in the 5′ tertile part of the *NF1* gene, especially loss-of-function variants. We are yet to confirm this correlation with a large dataset, but nonetheless, this is an interesting observation worth further analysis. Null variants within the initial portion of the protein are more prone to result in non-functional proteins, thus future studies should also involve the analysis of the potential mechanisms by which different proteins coded by variants located in different regions of the *NF1* gene lead to short stature in this at-risk population.

Despite a significant effort, only a few clear correlations were observed between particular pathogenic NF1 alleles and consistent clinical phenotypes. Whole-gene deletion of *NF1* leads to more severe cutaneous neurofibromas and neurological conditions and dysmorphic facial features [21]. A 3-bp in-frame deletion of exon 17 is associated with the typical pigmentary features of NF1 but not to cutaneous or surface plexiform neurofibromas [22]. One patient with this same in-frame deletion variant in our cohort exhibited only typical CALMs at the age of two. We also detected two individuals with the Arg1809Cys variant, which is known to be associated with multiple CALMs, learning disabilities, short stature, and pulmonic stenosis and to the absence of cutaneous neurofibromas or clinically apparent plexiform neurofibromas [23]. We examined these two patients. One was a 13-year-old boy with typical CALMs, astigmia, ADHD, and short stature, while the other is a two-year-old boy with typical CALMs and delayed language development. Cutaneous or plexiform neurofibromas were absent in both patients. This result showed that this correlation is valid also in Chinese NF1 patients.

## 5. Conclusions

We molecularly confirmed 68 individuals with NF1 out of 95 Chinese children with clinical suspicion of NF1 (a diagnostic yield of 71.6%) based mainly on the presence of multiple CALMs. NF1 is mainly diagnosed on the basis of clinical evaluation, but genetic testing is still helpful in early diagnosis, elucidating possible genotype–phenotype correlations and individualizing the follow-up according to the risk expected for each patient. Our study uncovered a significant number of novel NF1 pathogenic variants, indicating that it is meaningful to study patients of other ethnic backgrounds. We expect that understanding the mutational range in Chinese people with NF1 may reveal additional correlations between genotype and phenotype as well as treatment responses in the future. Our unique findings of a higher-than-expected de novo mutation rate and a possible correlation between short stature and variant location are to be further studied.

## Figures and Tables

**Figure 1 genes-10-00847-f001:**
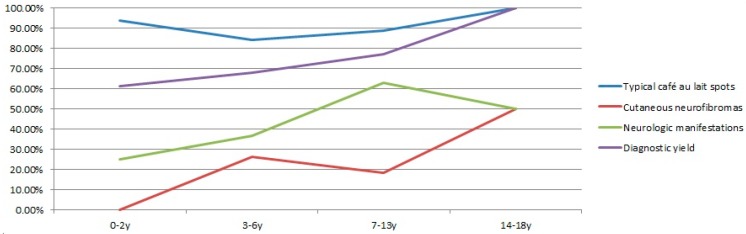
Correlations of the clinical expression and diagnostic yield for four age groups (infants and toddlers at age 0–2, preschooler at age 3–6, school-age children at age 7–13, and adolescents at age 14–17).

**Figure 2 genes-10-00847-f002:**
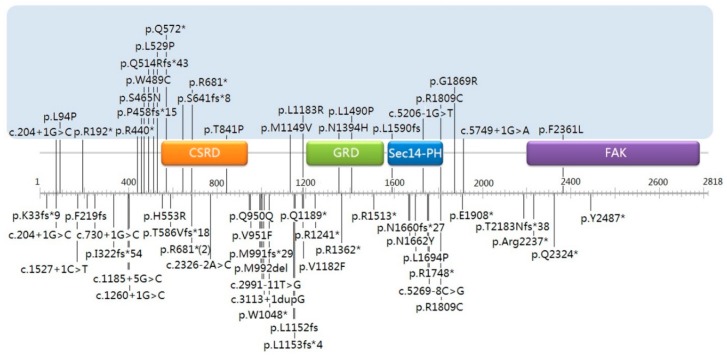
Variants’ distribution in patients with (above) or without (below) short stature.

**Table 1 genes-10-00847-t001:** Novel variants detected in Chinese neurofibromatosis 1 (NF1) patients.

Novel Variants			
de Novo		Inherited	
Null Variants	Other Variants	Null Variants	Other Variants
c.98_104delAAGTCAG p.K33fs*9	c.758_760delTGG p.V253del	c.1373delC p.P458fs*15	c.281T > C p.L94P
c.964delA p.I322fs*54	c.2521A > C p.T841P	c.1920dupC p.S641fs*8	c.1394G > A p.S465N
c.1527 + 1C > T	c.2850G > A p.Q950Q	c.2326-2A > C	c.3875A > G p.Y1292C
c.3113 + 1dupG	c.4984A > T p.N1662Y		
c.4770_4777delAAGTATTT p.L1590fs	c.4180A > C p.N1394H		
c.4950_4978dup p.N1660fs*27	c.5081T > C p.L1694P		
c.5206-1G > T	c.5269-8C > G		

**Table 2 genes-10-00847-t002:** Clinical profile of patients with or without pathogenic *NF1* variants.

NF1 Related Features	Patients with Pathogenic *NF1* Variants		Patients without Pathogenic *NF1* Variants	
Café-au-lait macules at birth	86.76%	(59/68)	33.33%	(9/27)
More than six café au lait macules	94.11%	(64/68)	70.37%	(19/27)
Cutaneous neurofibromas	19.11%	(13/68)	2.94%	(2/27)
Plexiform neurofibromas	1.47%	(1/68)	0%	(0/27)
Ocular findings	23.53%	(16/68)	18.52%	(5/27)
Neurologic manifestations	45.59%	(31/68)	11.11%	(3/27)
Musculoskeletal features	10.29%	(7/68)	1.47%	(1/27)

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
