# Peer review of "Clinical Presentation and Novel Pathogenic Variants among 68 Chinese Neurofibromatosis 1 Children"

_genes, 2019, doi:10.3390/genes10110847_

Round 1

Reviewer 1 Report

1.      In the introduction section, lines 44-46, the sentence “The diagnostic yield of NF1 is very high for adult and for those who met the clinical diagnostic criteria, the yield is somewhat lower for children who had not presented with a typical pattern of NF1 features” - the authors mean children who had not presented with a typical pattern of NF1 but are suspicious of NF1 by presenting café-au-lait macules or familial inheritance? The sentence needs to be rewritten.

2.      In the results section, lines 83-84, the authors state that “The age of the patients ranged from newborn to 17 years old with an average age of 6 years and 2 months”, but in other sections (Abstract, Results) they mention that Chinese children were recruited. Individuals above 14 years of age are not considered children.

3.      In line 85, the authors state that “Pathogenic or likely pathogenic variants were detected in 71.6% of patients (68/95)”, which is different from the abstract section “Pathogenic NF1 variants were detected in 71.6% (68/95) of patients”. Please correct.

4.      Can the authors comment on their overall yield for NF1 (71.6%), which is lower than other published estimates? Why do they think this is? Could any of the negative NF1 patients have Legius syndrome or other? What about mosaicism? It would be very interesting to know whether the sequencing approach is suitable for identifying mosaic NF1 mutations. How far into the NF1 introns did they sequence? What about copy-number assessment? The absence of exon dosage testing unfortunately means that large deletion/duplication mutations have not been sought and thus detected, so this is a failing although the authors briefly discuss this.  

5.      The authors should discuss the recruitment criteria - why do they use the presence of multiple café-au-lait macules and not the established clinical diagnosis criteria? Do they expect that the overall yield would increase if other criteria were used?

6.      Why do the authors use medical exome sequencing to sequence NF1 patients? Is there any other mutation detected in other genes that could explain the negative results? This would be valuable for discussion. Were the negative screening cases de novo or inherited cases?

7.      Novel splice site variants (c.5269-8C>G and c.2326-2A>C) should be defined. Were they classified as null variants according to ACMG criteria?

8.      The bioinformatics section should include more details regarding the variant selection, annotation and classification. Was the coverage analysis performed?

9.      Were the mutations described in this study searched in databases such as ClinVar or HGMD?

10.  All mutations described in the present study should be deposited in a mutation database (e.g. LOVD).

11.  Please provide the ACMG scores in the supplementary table. Mutations in Table 1 and supplementary table should be listed in order according to mutation position.

12.  Why do the authors try to correlate short stature with NF1 mutation position? What about other NF1 clinical characteristics?

13.  The finding that variants in the C-terminal region of NF1 gene are less likely to be associated with short stature is very interesting. However, no statistical comparison by type of mutation was performed. Also, no statistical comparisons were made to conclude that male patients seem more likely to have neurological problems compared with female patients.

14.  The authors found that 56.5% (13/23) of individuals exhibited short stature when their pathogenic variants are located in the 5’ tertile part of NF1 gene – this finding could be related to the severity of mutations located in the 5´position (null variants in the initial portion of the protein are more prone to result in non-functional proteins).

15.  There are a few spelling mistakes, double or missing spaces and grammatical errors in the manuscript. The text would benefit from additional proof-reading. I have included some errors here:

a.       Line 86 page 2. “12 splicing variants” appear twice in the text.

b.      Line 96 page 3.  “indicating that”

c.       Table 2. “Café au lait marcules at birth”

d.      Line 133 page 4. “start stature”

e.       Please review the sentence “Whether or not the higher de novo rate is ethnic group specific will need to be studied with a large dataset”

f.       Lines 193-194.  molecular confirmed

g.       Please review the sentence “The lowest yield was at 61.54% (16/26) for the infant and toddlers group. It improved to 67.86% and 77.14% for the preschooler and school-age groups respectively. Six adolescence patients were all molecular confirmed”

Author Response

In the introduction section, lines 44-46, the sentence “The diagnostic yield of NF1 is very high for adult and for those who met the clinical diagnostic criteria, the yield is somewhat lower for children who had not presented with a typical pattern of NF1 features” - the authors mean children who had not presented with a typical pattern of NF1 but are suspicious of NF1 by presenting café-au-lait macules or familial inheritance? The sentence needs to be rewritten.

Response: We have revised this sentence as ‘The diagnostic yield of NF1 is very high for adult and for those who met the clinical diagnostic criteria, the yield is somewhat lower for children who had not presented with a typical pattern of NF1 but are suspicious of NF1 by presenting café-au-lait macules.‘

In the results section, lines 83-84, the authors state that “The age of the patients ranged from newborn to 17 years old with an average age of 6 years and 2 months”, but in other sections (Abstract, Results) they mention that Chinese children were recruited. Individuals above 14 years of age are not considered children.

Response: By definition, children are those below the age of 18.

In line 85, the authors state that “Pathogenic or likely pathogenic variants were detected in 71.6% of patients (68/95)”, which is different from the abstract section “Pathogenic NF1 variants were detected in 71.6% (68/95) of patients”. Please correct.

Response: We thank reviewer pointing out this discrepancy. We have revised the abstract section.

Can the authors comment on their overall yield for NF1 (71.6%), which is lower than other published estimates? Why do they think this is? Could any of the negative NF1 patients have Legius syndrome or other? What about mosaicism? It would be very interesting to know whether the sequencing approach is suitable for identifying mosaic NF1 mutations. How far into the NF1 introns did they sequence? What about copy-number assessment? The absence of exon dosage testing unfortunately means that large deletion/duplication mutations have not been sought and thus detected, so this is a failing although the authors briefly discuss this.

Response: We agree with the reviewer that this is important point to address. In addition to what was discussed in the manuscript, we now have added new information the reviewer had prompted us to include. We have updated the discussion. " Although the medical exome approach we used can help differential diagnosis of NF2-like syndrome such as Legius syndrome and detection of mosaic variants at some levels, the missed detections of small deletions, low-level mosaic mutations and deep-intronic variants contributed to our overall (71.6%) lower than typical level of diagnostic yield. But this yield also reflects the uncertainty of NF1 clinical diagnosis among children."

Variants in SPRED1 leading to Leguis syndrome are routinely screened during our medical exome sequencing. Our pipeline is capable to detecting mosaic variants at 15% level or above. NGS based test will miss low-level mosaicism. The target capture only extends ±20bp in introns and is unable to detect deep intronic variants.

The authors should discuss the recruitment criteria - why do they use the presence of multiple café-au-lait macules and not the established clinical diagnosis criteria? Do they expect that the overall yield would increase if other criteria were used?

Response: The established clinical diagnosis criteria are great for adult and older children but are not perfect for young children since the manifestation of additional features are age-dependent. Early diagnosis become possible with molecular testing for children with NF1 at young ages while only presenting with café-au-lait macules (NF1 as a major differential diagnosis for CALM). Our data reflect the real-world situation when patients come to seek medical attention with what they have at these ages. We have revised the introduction to make this point better.

Why do the authors use medical exome sequencing to sequence NF1 patients? Is there any other mutation detected in other genes that could explain the negative results? This would be valuable for discussion. Were the negative screening cases de novo or inherited cases?

Response: Medical exome sequencing is a cost-effective choice we have demonstrated for us in China (please see our previous publication in Genetics in Medicine PMID: 29095814). Since NF1 is a large gene and single gene test has no advantage than testing the whole medical exome. Plus, as the reviewer correctly pointed out, medical exome sequencing offer possibility of detecting other causal variants. But so far we have not detected any causal variants for these NF1-negative cases.  

Novel splice site variants (c.5269-8C>G and c.2326-2A>C) should be defined. Were they classified as null variants according to ACMG criteria?

Response: c.2326-2A>C can be classified as null variants since this is a canonical splicing variant. We need experimental evidence to confirm the splicing effect of c.5269-8C>G, but since this is a de novo variant and the existing evidence already support this as a likely pathogenic variant. We have revised the table.

The bioinformatics section should include more details regarding the variant selection, annotation and classification. Was the coverage analysis performed?

Response: Annotations were done automatically with two commercial software IVA and TGex. All the exonic and splicing variants including possible mosaicism variants in genes with related disorders were selected and classified. Routine quality control of coverage and sequencing depth were performed.

Were the mutations described in this study searched in databases such as ClinVar or HGMD?

Response: Yes, all the mutations identified were searched in HGMD, Clinvar and Mastermind Genomic Search Engine (https://www.genomenon.com/mastermind).

All mutations described in the present study should be deposited in a mutation database (e.g. LOVD).

Response: We will submit those variants to Clinvar database once we granted authorization from our organization.

Please provide the ACMG scores in the supplementary table. Mutations in Table 1 and supplementary table should be listed in order according to mutation position.

Response: We have added ACMG scores and realigned those variants in table.

Why do the authors try to correlate short stature with NF1 mutation position? What about other NF1 clinical characteristics?

Response: We did correlation study for other features but height information is the most readily available one for every patient. For all clinical features, we only identified this possible correlation.

The finding that variants in the C-terminal region of NF1 gene are less likely to be associated with short stature is very interesting. However, no statistical comparison by type of mutation was performed. Also, no statistical comparisons were made to conclude that male patients seem more likely to have neurological problems compared with female patients.

Response; We agree with reviewer that our findings is of interest but far from convincing. They all need further rigorous validation. We will work on this with larger dataset and with statistical analysis. 

The authors found that 56.5% (13/23) of individuals exhibited short stature when their pathogenic variants are located in the 5’ tertile part of NF1 gene – this finding could be related to the severity of mutations located in the 5´position (null variants in the initial portion of the protein are more prone to result in non-functional proteins).

Response: We agree with reviewer that this is a possibility. We have added this point in the discussion.

There are a few spelling mistakes, double or missing spaces and grammatical errors in the manuscript. The text would benefit from additional proof-reading. I have included some errors here:

Line 86 page 2. “12 splicing variants” appear twice in the text.

Line 96 page 3. “indicating that”

Table 2. “Café au laitmarcules at birth”

Line 133 page 4. “start stature”

Please review the sentence “Whether or not the higher de novo rate is ethnic group specific will need to be studied with a large dataset”

Lines 193-194. molecular confirmed

Please review the sentence “The lowest yield was at 61.54% (16/26) for the infant and toddlers group. It improved to 67.86% and 77.14% for the preschooler and school-age groups respectively. Six adolescence patients were all molecular confirmed”

Response: We really appreciate reviewer's meticulous correction. We have revised the manuscript accordingly.

Reviewer 2 Report

The study is interesting and well written. However some major and minor improvements may be done:

1) In the Introduction section, it should be mentioned the belonging of NF1 to the so called RASopathies. Such clarification could help the reader to better undertsand molecular and pathophysiological basis of NF1, and then its clinical features. In the same section more recent references about NF1 incidence (documenting values higher than 1:3000) should be added;

2) in the Introduction section should be added that molecular testing in NF1 is helpful other than to confirm the diagnosis early in life, also to rule out other rasopathies/diseases sharing overlapping features;

3) in the Patients section, the months of the years 2017 and 2019 should be added;

4) in the Genetic findings subparagraph, instead of "newborn", the days of life of the proband should be specified; in the same section, "12 splicing variants" is repeated twice, and in the same line (87) "two" could be changed in 2 (according to the numbering used for the other mutations);

4)line 97, "indicating that clinical diagnosis" should be used instead of "indicating the clinical...", and line 100, "were all confirmed by molecular testing" instead of "molecular confirmed";

5) in the Clincal characteristics section, the acronym CALMs should be used when appropriate; in the same paragraph, the ocular involvement (in light of its important role in the phenotype of NF1) should be assessed, as well as the musculoskeletal features and neuroradiological findings, that should be better described;

6)line 134, "start" should be changed in "short" stature;

7)line 161, "that lead to" should be modified in "involved in";

8)lines 186-187, the sentence "what was described for" should be removed, and line 188 "than" shuld be used instead of "of that in the";

9) the Authors have advanced a possible correlation between variants at the 5' tertile part of NF1 and short statureare they able to better elucidate this point and to describe if other possible correlations could be observed (i.e. between other clinical manifestations as skeletal abnormalities/tumors, and specific types of mutations)?;

10)line 212, a language deficit may be diagnosed after two years of age. The Authors should clarify this point;

11) it could be useful to move the sentence (after having shortened it) going from line 202 to line 214, to the conclusions section. In this latter one, the Authors should include a phrase which underlies that diagnosis of NF1 is based on clinical evaluation, when NIH criteria are present, and that however identification of pathologic variants is important, not only to understand genotype-phenotype correlations, but also to individualize the follow-up according to the risk expected for each patient

Author Response

The study is interesting and well written. However some major and minor improvements may be done:

In the Introduction section, it should be mentioned the belonging of NF1 to the so called RASopathies. Such clarification could help the reader to better undertsand molecular and pathophysiological basis of NF1, and then its clinical features. In the same section more recent references about NF1 incidence (documenting values higher than 1:3000) should be added;

We have revised the introduction part to including these two points.

in the Introduction section should be added that molecular testing in NF1 is helpful other than to confirm the diagnosis early in life, also to rule out other rasopathies/diseases sharing overlapping features;

Response: We thank reviewer's great input and have revised the manuscript to include this point.

in the Patients section, the months of the years 2017 and 2019 should be added;

Response: We have revised this section accordingly.

in the Genetic findings subparagraph, instead of "newborn", the days of life of the proband should be specified; in the same section, "12 splicing variants" is repeated twice, and in the same line (87) "two" could be changed in 2 (according to the numbering used for the other mutations);

Response: We have revised the paragraph accordingly.

line 97, "indicating that clinical diagnosis" should be used instead of "indicating the clinical...", and line 100, "were all confirmed by molecular testing" instead of "molecular confirmed";

Response: We appreciate reviewer's suggestions. We have revised these sentences accordingly.

in the Clincal characteristics section, the acronym CALMs should be used when appropriate; in the same paragraph, the ocular involvement (in light of its important role in the phenotype of NF1) should be assessed, as well as the musculoskeletal features and neuroradiological findings, that should be better described;

Response: We have revised the whole manuscript using the acronym CALMs. In this cohort, patients' ocular and musculoskeletal phenotypes were not well assessed and this is a short come of this study.  

line 134, "start" should be changed in "short" stature;

Response: Thanks for pointing out this. We have corrected the typo.

line 161, "that lead to" should be modified in "involved in";

Response: We have revised it according to the suggestion.

lines 186-187, the sentence "what was described for" should be removed, and line 188 "than" shuld be used instead of "of that in the";

Response: We have revised it according to the suggestion.

the Authors have advanced a possible correlation between variants at the 5' tertile part of NF1 and short stature; are they able to better elucidate this point and to describe if other possible correlations could be observed (i.e. between other clinical manifestations as skeletal abnormalities/tumors, and specific types of mutations)?;

Response: We did correlation study for other features but height information is the most readily available one for every patient. For all clinical features, we only identified this possible correlation. We agree with reviewer that this is an interesting but preliminary finding requiring further validation and elucidation.

10)line 212, a language deficit may be diagnosed after two years of age. The Authors should clarify this point;

Response: We have revised this to ‘delayed language development’ to better reflect the situation.

11) it could be useful to move the sentence (after having shortened it) going from line 202 to line 214, to the conclusions section. In this latter one, the Authors should include a phrase which underlies that diagnosis of NF1 is based on clinical evaluation, when NIH criteria are present, and that however identification of pathologic variants is important, not only to understand genotype-phenotype correlations, but also to individualize the follow-up according to the risk expected for each patient

Response: We appreciate reviewer's advice. We have revised the conclusion section accordingly.

Reviewer 3 Report

In this study, “Clinical presentations and novel pathogenic variants among 68 Chinese neurofibromatosis 1 children”, Ruen et al describe the genetic and some clinical features of 95 Chinese children referred for evaluation based on the suspicion of NF1.  The age range of the children acceptes was newborn to age 17 and the primary reason for referral was the presence of multiple café-au-lait macules.  Exome and confirmation Sanger sequencing was performed on all 95 children.  They found that 68 out of 95 children had a pathologic or likely pathologic variant and conclude that Chinese NF1 patients are an understudied population-based on findings of 20 pathogenic variants not previously reported and a higher than expected rate of 77.9% de novo variants.  These are important observations on their own, however, the size of the data population as well as the inclusion of young childen allows for interesting observations about the presence of other clinical exam findings in early life such as cutaneous neurofibromas. What is not well justified or explained is the analysis of the C-terminal region versus N-terminal region varients and stature.  There are several critical flaws in this analysis: what was the age range included?  How was normal stature defined for each age group?  How was the variation across age group accounted for?  There is no longitudinal data to allow for analysis of change over time.  There are also multiple confounding variables that are not controlled for (size of the parents, sex, socioeconomic status).  Overall, however, the most important limitation to this analysis is the lack of a plausible biologic rationale for why such egional varients would regulate growth.  Finally, this is not a clinical question that impacts clinical care decisions and hence, is of low impact.  This reviewer’s suggestion is to eliminate this specific analysis (stature and N- versus C-terminus variant for all of the reasons listed above.   In place of this analysis, it would be very informative to have additional clinical data such as: number and distribution of cutaneous neurofibromas in each of their identified age groups (-02, 3-6, 7-13, 14-18) and to clarify that that data is based on visual inspection. It would also be helpful to have similar data about Lisch Nodules.  In terms of the “neurologic” manifestation, we would suggest sperating cognitive manifestations (ADHD, developmental delay, etc) uniquely from headache, vision changes, focal weakness or sensory changes.  Finally, it would be helpful to have these results reported by sex and age group. 

Regading genetic testing sensitivity, it would be helpful to know how many children had both a 1st degree relative with NF1 as well as CAL spots – these are please expected to have positive NF1 testing.  May be skewing the total number reported.

In addition, it would be very helpful to search for and report the frequency of mutations that have reported genotype-phenotype correlations in this specific population: 

c.2970-2972 delAAT (Upadhyaya et al, 2007)

proximal NF1‐REP‐a and distal NF1‐REP–c for the 1.4 Mb type‐1 microdeletion; SUZ12 and SUZ12P for the 1.2 Mb type‐2 microdeletion (Pasmant et al, 2010)

NF1 Codons 844–848 (Koczkowska et al, 2018)

After these extensive revisions are made, the discussion section will accordingly need revision.  A major opportunity is to hypothesize how understanding the mutational range in Chinese people with NF1 may influence disease severity (might it be that the lower than expected rate fo plexiform neurofibromas are related to specific mutational subtypes in Chinese people?) and response to treatment given that several NF1 plexiform neurofibroma treatments are on the horizon. 

In addition the the major changes suggested above, minor issues that should be addressed include:

- There are grammatical errors throughout the paper.

- Line 31 (in the abstract): “under-studies” should be “understudied” NF1 patient populations

As the authors point out, most patients did not receive an ophthalmological evaluation to look for Lisch nodules. This would be interesting to evaluate in future renditions.

- Table 2: macule is misspelled a “marcule”.

- line 113-114: when reporting only one person with plexiform neurofibroma, would be helpful to explain if MRI was performed or not; this number is lower than in published series.

- Line 134: space is needed between “of” and NF1.

- Figure 2 is informative but has variants clustered too close together which affect readability.

- Lines194-195: The phrase “molecular confirm NF1 children” needs to be modified to be grammatically correct.

Author Response

In this study, “Clinical presentations and novel pathogenic variants among 68 Chinese neurofibromatosis 1 children”, Ruen et al describe the genetic and some clinical features of 95 Chinese children referred for evaluation based on the suspicion of NF1.  The age range of the children acceptes was newborn to age 17 and the primary reason for referral was the presence of multiple café-au-lait macules.  Exome and confirmation Sanger sequencing was performed on all 95 children.  They found that 68 out of 95 children had a pathologic or likely pathologic variant and conclude that Chinese NF1 patients are an understudied population-based on findings of 20 pathogenic variants not previously reported and a higher than expected rate of 77.9% de novo variants.  These are important observations on their own, however, the size of the data population as well as the inclusion of young childen allows for interesting observations about the presence of other clinical exam findings in early life such as cutaneous neurofibromas.

What is not well justified or explained is the analysis of the C-terminal region versus N-terminal region varients and stature.  There are several critical flaws in this analysis: what was the age range included?  How was normal stature defined for each age group?  How was the variation across age group accounted for?  There is no longitudinal data to allow for analysis of change over time.  There are also multiple confounding variables that are not controlled for (size of the parents, sex, socioeconomic status).  Overall, however, the most important limitation to this analysis is the lack of a plausible biologic rationale for why such egionalvarients would regulate growth.  Finally, this is not a clinical question that impacts clinical care decisions and hence, is of low impact.  This reviewer’s suggestion is to eliminate this specific analysis (stature and N- versus C-terminus variant for all of the reasons listed above.   In place of this analysis, it would be very informative to have additional clinical data such as: number and distribution of cutaneous neurofibromas in each of their identified age groups (-02, 3-6, 7-13, 14-18) and to clarify that that data is based on visual inspection. It would also be helpful to have similar data about Lisch Nodules.  In terms of the “neurologic” manifestation, we would suggest sperating cognitive manifestations (ADHD, developmental delay, etc) uniquely from headache, vision changes, focal weakness or sensory changes.  Finally, it would be helpful to have these results reported by sex and age group.

Response: We agree with reviewer that there are many confounding variables in this analysis that were not accounted for. While the correlation between variant distribution and short stature is observational at this point, we think it phenomena is worth pointing out so additional studies can further (dis)validate this finding and biological meaning can be examined. The normal stature was defined using a Chinese population study as height is age, ethnicity and gender specific. Due to a relative small sample included in this study, we did not report the results for subgroups based on sex and age. The reviewer's points are all validate and these are our guide for further analysis.

Regading genetic testing sensitivity, it would be helpful to know how many children had both a 1st degree relative with NF1 as well as CAL spots – these are please expected to have positive NF1 testing.  May be skewing the total number reported.

Response: Indeed, we agree with reviewer that family history is a strong indication for NF1 genetic testing.  In this cohort all individuals with clear a family history and CALMs had positive NF1 testing results. We have added this in the manuscript now

In addition, it would be very helpful to search for and report the frequency of mutations that have reported genotype-phenotype correlations in this specific population:

c.2970-2972 delAAT (Upadhyaya et al, 2007)

proximal NF1‐REP‐a and distal NF1‐REP–c for the 1.4 Mb type‐1 microdeletion; SUZ12 and SUZ12P for the 1.2 Mb type‐2 microdeletion (Pasmant et al, 2010)

NF1 Codons 844–848 (Koczkowska et al, 2018)

Response: This is indeed very interesting to do and we did it. but it turns out, we only have one patient with those previously report variants with reported genotype-phenotype correlations (c.2970-2972 delAAT). And we have revised the last paragraph of the discussion to include this information.

After these extensive revisions are made, the discussion section will accordingly need revision.  A major opportunity is to hypothesize how understanding the mutational range in Chinese people with NF1 may influence disease severity (might it be that the lower than expected rate fo plexiform neurofibromas are related to specific mutational subtypes in Chinese people?) and response to treatment given that several NF1 plexiform neurofibroma treatments are on the horizon.

Response: We agree with reviewer's points. We have added this point to the discussion. Since many of these studies requires a larger sample size and follow-up information for prognosis, we hope to gather more data for a larger study for Chinese NF1 patients.

In addition the the major changes suggested above, minor issues that should be addressed include:

- There are grammatical errors throughout the paper.

Response: we regret for the grammatical errors and other reviewers had also pointed out. We have then correct in the revised version.

- Line 31 (in the abstract): “under-studies” should be “understudied” NF1 patient populations

Response: Thanks and it is fixed.

As the authors point out, most patients did not receive an ophthalmological evaluation to look for Lisch nodules. This would be interesting to evaluate in future renditions.

Response: We agree and it should be evaluated for every patient.

- Table 2: macule is misspelled a “marcule”.

Response: Thanks and it is fixed.

- line 113-114: when reporting only one person with plexiform neurofibroma, would be helpful to explain if MRI was performed or not; this number is lower than in published series.

Response: MRI was not routinely performed. It is classified in the revised version.

- Line 134: space is needed between “of” and NF1.

- Figure 2 is informative but has variants clustered too close together which affect readability.

- Lines194-195: The phrase “molecular confirm NF1 children” needs to be modified to be grammatically correct.

Response: Thanks and it is fixed. Readability of figure2 will be improved when printed in larger size while production.

Reviewer 4 Report

In this manuscript Yao and colleagues used exome sequencing to analyze for mutations 95 young patients suspected of having neurofibromatosis type 1, based on the presence of CaLS with or without additional findings.

The manuscript could be of interest for the people involved in NF1 but needs major revision.

Comments

- Title:

I suggest to write: Clinical presentation ……” instead of “Clinical presentations……..”

Introduction:

- please acknowledge previous works focused on dissecting the mutation spectrum and clinical features of NF1 Chinese population: i.e. Chai et al., BMC Med Genet. 2019 Sep 18;20(1):158; Zhang et al., Sci Rep. 2015 Jun 9;5:11291; Lee et al., Hum Mutat. 2006 Aug;27(8):832.

Materials and Methods

Patients

- Please, provide a more detailed clinical description of the selected cohort: age (median, max, min), family history, gender, clinical features. The brief description provided at the beginning of the results section (page 3, line 96) could be moved to the patients’ section.

- Page 2, line 61: the authors reported that “Individuals with clinical suspicion of NF1, mostly due to the presence of multiple CaLMs, or with close relatives previously diagnosed with NF1 were referred…… Familiarity for neurofibromatosis type 1 in the absence of any NF1-related clinical sign (i.e. at least CaLS) is generally not considered an indication for mutation testing. Analysis is generally performed on the proband and subsequently to identification of the familial mutation extended to other family members. Did the authors used exome sequencing to analyse non-affected individuals with familiarity for NF1? I suggest the authors to explain the reasons of this approach since I expect not many mutations to be identified using such approach. Neurofibromatosis type 1 is a progressive disorder but generally CaLMs appear very early. I suggest patients without any NF1 clinical related sign to be excluded from the analysis.

Page 2, line 64: “Medical exome…..; I suggest moving this part to the 2.2 molecular genetics section.

Molecular genetics section

Page 2, line 71. The authors used medical exome sequencing to analyze their patients. Did they use specific filtering criteria (i.e. analysis of selected genes) or they read the entire exome? Considering that neurofibromatosis type 1 is a single gene disorder, with phenotypic overlapping with few other conditions, i.e. Legius syndrome due to SPRED1 mutations, or Noonan syndrome with multiple lentigines patients at young age due to PTPN11 specific mutations, is there a specific motivation for analysing the entire exome? Some explanation is provided in the discussion section, but I suggest a better description of the mutation analysis approach (analyzed genes, etc.) to be provided in the methods section.

Results

Genetic findings

- I suggest the authors to verity the numbers reported in this section. The authors state that 68 mutations were identified (line 98); 53 mutations were de novo, 9 were maternally inherited and 5 were inherited paternally. 53+9+5=67. Please check also the percentages.

- Figure 1. The authors found that diagnostic yield increased with the increasing of age at diagnosis. This is expected in NF1 because of the progressive nature of the disease. I found the figure difficult to interpret. The authors may want to add numbers (how many patients were analyzed for each group). The authors may want to think of describing the clinical characteristics of their patients at different age in a table below the figure, or alternatively using histograms. Please, specify in the patients section what is intended for neurologic manifestations, or musculoskeletal features.

- variant distribution and short stature paragraph. I suggest to provide statistical evidence of the association between position and/or type of NF1 variants along the coding sequence and stature, if possible.

Discussion

- Page 5, line 200; Legius syndrome is not considered an NF2-like syndrome but an NF1-like syndrome; schwannomatosis is considered more an NF2-related syndrome. Generally, exome analysis detect small indels, but is less sensible in detecting CNVs, either intragenic or including the entire gene that together represent more than 5% of the NF1 mutation spectrum. The authors may want to include these gene anomalies among those that were possibly missed by exome analysis.

Minor comments

- Introduction: page 1, line 40: and axillary and inguinal freckling is repeated twice.

- Many times CaLMs is written CLAMs

Author Response

Comments

- Title:

I suggest to write: Clinical presentation ……” instead of “Clinical presentations……..”

Response: We now used “Clinical presentation” in our title.

Introduction:

- please acknowledge previous works focused on dissecting the mutation spectrum and clinical features of NF1 Chinese population: i.e. Chai et al., BMC Med Genet. 2019 Sep 18;20(1):158; Zhang et al., Sci Rep. 2015 Jun 9;5:11291; Lee et al., Hum Mutat. 2006 Aug;27(8):832.

Response: We thank reviewer for pointing out the new and previous publications on this topic. We have now referenced these previous works.

Materials and Methods

Patients

- Please, provide a more detailed clinical description of the selected cohort: age (median, max, min), family history, gender, clinical features. The brief description provided at the beginning of the results section (page 3, line 96) could be moved to the patients’ section.

Response: The detailed description of our patients are provided in the supplemental material. We have moved the brief description from the results section to the patients’ section.

- Page 2, line 61: the authors reported that “Individuals with clinical suspicion of NF1, mostly due to the presence of multiple CaLMs, or with close relatives previously diagnosed with NF1 were referred…… Familiarity for neurofibromatosis type 1 in the absence of any NF1-related clinical sign (i.e. at least CaLS) is generally not considered an indication for mutation testing. Analysis is generally performed on the proband and subsequently to identification of the familial mutation extended to other family members. Did the authors used exome sequencing to analyse non-affected individuals with familiarity for NF1? I suggest the authors to explain the reasons of this approach since I expect not many mutations to be identified using such approach. Neurofibromatosis type 1 is a progressive disorder but generally CaLMs appear very early. I suggest patients without any NF1 clinical related sign to be excluded from the analysis.

Response: We may have confused the reviewer and the readers. The patients we included in our study were patients with CALMs or those who have been clinically diagnosed as NF1 without molecular confirmation. And some of them do have close relatives diagnosed as NF1. We didn’t test non-affected individuals with familiarity for NF1. We have revised this sentence to avoid misunderstanding.

Page 2, line 64: “Medical exome…..; I suggest moving this part to the 2.2 molecular genetics section.

Response: We have revised the manuscript accordingly.

Molecular genetics section

Page 2, line 71. The authors used medical exome sequencing to analyze their patients. Did they use specific filtering criteria (i.e. analysis of selected genes) or they read the entire exome? Considering that neurofibromatosis type 1 is a single gene disorder, with phenotypic overlapping with few other conditions, i.e. Legius syndrome due to SPRED1 mutations, or Noonan syndrome with multiple lentigines patients at young age due to PTPN11 specific mutations, is there a specific motivation for analysing the entire exome? Some explanation is provided in the discussion section, but I suggest a better description of the mutation analysis approach (analyzed genes, etc.) to be provided in the methods section.

Response:Analysis of exome-sequencing data of our patients are indeed the critical part of our study. We focused on NF1 first, but also pay attention to all related genes including Rasopathy related genes like PTPN11, genes related with CALMs like FANCA, GNAS, genes related with conditions need differential diagnosis like SPRED1. Actually the entire exome sequencing data were analyzed to avoid any other possibly unexpected conditions. We have revised this section to make it clear we have analyzed the variants from the entire exome.

Results

Genetic findings

- I suggest the authors to verity the numbers reported in this section. The authors state that 68 mutations were identified (line 98); 53 mutations were de novo, 9 were maternally inherited and 5 were inherited paternally. 53+9+5=67. Please check also the percentages.

Response: We appreciate reviewer's carefulness, we regret for the error. Indeed, 54 variants were de novo and we have confirmed that these percentages are correct.

- Figure 1. The authors found that diagnostic yield increased with the increasing of age at diagnosis. This is expected in NF1 because of the progressive nature of the disease. I found the figure difficult to interpret. The authors may want to add numbers (how many patients were analyzed for each group). The authors may want to think of describing the clinical characteristics of their patients at different age in a table below the figure, or alternatively using histograms. Please, specify in the patients section what is intended for neurologic manifestations, or musculoskeletal features.

Response: We have revised the description of figure 1 adding number of patients analyzed for each group. Detailed clinical characteristics of patients have been provided in the supplemental table 1. Detailed explanation for those clinical conditions has been revised in the results section. “……neurological manifestations which included attention deficit and hyperactivity (ADHD), developmental delay, language deficit, sleep disorders, learning problems, intellectual disability and headache of unknown cause.” Musculoskeletal problems were rare and non-uniform, so we didn’t list them in the results, but were all recorded in the supplemental table.

- variant distribution and short stature paragraph. I suggest to provide statistical evidence of the association between position and/or type of NF1 variants along the coding sequence and stature, if possible.

Response: Although we have genetically diagnosed 68 patients and analyzed their clinical features to find a possible genotype-phenotype correlation, yet the numbers of patients in each group are still limited, the statistical analysis for significance has limited power. We opt to further conduct such analysis with a larger dataset in our next phase of the study.  

Discussion

- Page 5, line 200; Legius syndrome is not considered an NF2-like syndrome but an NF1-like syndrome; schwannomatosis is considered more an NF2-related syndrome. Generally, exome analysis detect small indels, but is less sensible in detecting CNVs, either intragenic or including the entire gene that together represent more than 5% of the NF1 mutation spectrum. The authors may want to include these gene anomalies among those that were possibly missed by exome analysis.

Response: We agree with reviewer that small copy number variants involving NF1 would be missed by exome sequencing but not large deletions encompassing NF1 and neighboring genes. And this may be one factor for the lower than expected diagnostic yield in our study as we stated in the discussion. A more comprehensive genetic testing strategy including intragenic CNV detection should be considered in our future study.

Minor comments

- Introduction: page 1, line 40: and axillary and inguinal freckling is repeated twice.

- Many times CaLMs is written CLAMs

Response: We thank reviewer for pointing out the spelling errors, we have fixed them in the revision.

Reviewer 5 Report

The authors provide a review of their series of patients with NF1 and molecular findings.  The primary concern about the manuscript is that it is not novel and provides very little new information not already available in the medical literature and hence think that it is of low priority for publication.

Comments:

1.The NF1 gene is a very large gene and hence the fact that the authors identified many novel variants is not unusual nor particularly informative.

2.  The number of NF1 individuals they report is not large compared to many other reports and laboratories.

3.  It appears that exome sequencing was used and hence one questions if they evaluated for whole gene deletions which are know to cause a portion of cases of NF1.

4.  The average age of the patients was young at 6 years of age and hence makes comparison of genotype-phenotype relations imperfect as the findings of NF1 are age related and most individuals at this age will not have many clinical features.

5.  Most individuals did not have ophthalmology evaluations so there is significant missing information.

6.  It is not clear if they ruled out other conditions such as Legius syndrome for those with just the pigmentary findings.

Author Response

Comments:

The NF1 gene is a very large gene and hence the fact that the authors identified many novel variants is not unusual nor particularly informative.

Response: We agree that identification of novel variants in larger genes such as NF1 is not surprising, particularly in an understudied population. Nevertheless, the findings would enrich the resources for genotype-phenotype correlation studies on a global scale.

The number of NF1 individuals they report is not large compared to many other reports and laboratories.

Response: This is one of the large cohorts of Chinese NF1 patients and we are accumulating more cases. The findings is less a competition for cohort size but rather for potential genotype-phenotype correlations based on data from an independent population. Certainly we wish to use much a large dataset for further study.  

It appears that exome sequencing was used and hence one questions if they evaluated for whole gene deletions which are know to cause a portion of cases of NF1.

Response; We agree that CNVs, either intragenic or large deletion involving the entire NF1 gene accounted for more than 5% of the NF1 pathogenic variants. Our exome sequencing analysis would miss small intragenic copy number variants but will be able to detect large deletions encompassing NF1 and neighboring genes. For genotype-phenotype analysis, we mainly focused on small variants which are the variants we detected in this cohort.

The average age of the patients was young at 6 years of age and hence makes comparison of genotype-phenotype relations imperfect as the findings of NF1 are age related and most individuals at this age will not have many clinical features.

Response: We have separated our cohort into four different age groups and investigated the presence of different clinical features and their diagnostic yield. We also tried to find genotype-phenotype relations that could be observed from young patients like short stature. We think it is important to study age-dependent correlations

Most individuals did not have ophthalmology evaluations so there is significant missing information.

Response: We agree this is a big missing information. We will schedule ophthalmology evaluations for the cohort during their future follow-up visits. This is an on-going study. We thank reviewer for the constructive comments and suggestions.

It is not clear if they ruled out other conditions such as Legius syndrome for those with just the pigmentary findings.

Response: Leguis syndromes and other related conditions were inspected both clinically and molecularly. Regarding exome sequencing data analysis, we focused on NF1 first, but also pay attention to all related genes including Rasopathy related genes like PTPN11, genes related with CALMs like FANCA, GNAS, genes related with conditions need differential diagnosis like SPRED1. Actually the entire exome sequencing data were analyzed to avoid any other possibly unexpected conditions. We have revised this section to make it clear we have analyzed the variants from the entire exome.

Round 2

Reviewer 1 Report

Abbreviation of café-au-lait macules is mispelled in some sections of the new version of the manuscript (CLAMs).

Author Response

Response: We thank the reviewer for pointing out the misspelled abbreviation, we have corrected the errors in this revised version.

Reviewer 4 Report

Please, double check text for typos. No additional comments.

Reviewer 5 Report

The authors provided appropriate responses but it does not change the significance of the information as there were little changes to the actual content of the manuscript.  The current manuscript provides only a minimal increase to the medical literature.